# Insights into the Functional Components in Wheat Grain: Spatial Pattern, Underlying Mechanism and Cultivation Regulation

**DOI:** 10.3390/plants12112192

**Published:** 2023-05-31

**Authors:** Yingxin Zhong, Yuhua Chen, Mingsheng Pan, Hengtong Wang, Jiayu Sun, Yang Chen, Jian Cai, Qin Zhou, Xiao Wang, Dong Jiang

**Affiliations:** College of Agriculture, Nanjing Agricultural University, Nanjing 210095, China; yingxin-zhong@njau.edu.cn (Y.Z.); 2019101010@njau.edu.cn (Y.C.); 2022201097@stu.njau.edu.cn (M.P.); 2021101027@stu.njau.edu.cn (H.W.); 2022101071@stu.njau.edu.cn (J.S.); 11119319@njau.edu.cn (Y.C.); caijian@njau.edu.cn (J.C.); qinzhou@njau.edu.cn (Q.Z.); xiaowang@njau.edu.cn (X.W.)

**Keywords:** quality, grain, spatial pattern, wheat, endosperm, underlying mechanism, cultivation regulation, research progress

## Abstract

Wheat is a staple crop; its production must achieve both high yield and good quality due to worldwide demands for food security and better quality of life. It has been found that the grain qualities vary greatly within the different layers of wheat kernels. In this paper, the spatial distributions of protein and its components, starch, dietary fiber, and microelements are summarized in detail. The underlying mechanisms regarding the formation of protein and starch, as well as spatial distribution, are discussed from the views of substrate supply and the protein and starch synthesis capacity. The regulating effects of cultivation practices on gradients in composition are identified. Finally, breakthrough solutions for exploring the underlying mechanisms of the spatial gradients of functional components are presented. This paper will provide research perspectives for producing wheat that is both high in yield and of good quality.

## 1. Introduction

Wheat is arguably the most important food crop in the world. It is unique for the role it plays in a variety of processes, including the creation of leavened and unleavened breads, pasta, noodles, and pastries [1]. Its functioning is markedly different from that of other grains. There are many quality classes of wheat grown around the world and each class of wheat is used for specific processed products [2]. Different grading standards have been established with respect to the various wheat classes produced in these countries, classified on the basis of their end-use characteristics (Table 1).

The wheat kernel comprises endosperm, embryo, and bran, with endosperm accounting for around 82%, embryo accounting for approximately 3%, and bran accounting for approximately 15% of the total [3]. Bran is made up of the epicarp, endocarp, testa, nucellar epidermis, and an aleurone layer, with the aleurone layer being the outermost cell that surrounds the endosperm and accounts for around 50% of the bran’s weight [4,5,6]. Starch, which accounts for around 70% of the grain weight and gives energy to the human body, is the chemical component comprising the largest percentage of the wheat grain, followed by protein, which accounts for approximately 8–20% [7]. Both starch and protein determine the yield and quality of wheat grains to a considerable degree. Furthermore, the grain has a variety of nutrients that are beneficial to the human body, such as dietary fiber, trace minerals, plant bioactivity nutrients, and carotenoids [8]. Differences in composition between these cell types have been known for many years, with the sub-aleurone cells being richer in protein and with fewer starch granules, which are less regular in shape compared with the other starchy endosperm cells [9]. More information on variations in composition within the starchy endosperm has come from studies using two approaches: cross-section staining combined with microscopy (Figure 1) and the progressive removal of layers from the outside of the grain by pearling (Figure 2). Both approaches exhibited spatial heterogeneity of the protein and starch in grain.

It is well known that the protein content in wheat grain can be regulated by nitrogen by up to 2 times [10]. It has been reported that spatial distribution can also be modified by cultivation practices, including nitrogen supplementation, planting density, regulator spraying, and so on [11,12,13]. What cannot be ignored is that when using milling technology, the outer 20–30% of the grain, including the pericarp, seed coat, aleurone, embryo, and part of the outer endosperm will be removed to produce fine flour. Thus, cultivation practices govern the final commercial flour quality by regulating the spatial distribution of grain components, and it is critical to understand the regulatory impacts of different cultivation practices on the spatial distribution of grain quality.

A topical overview regarding the study of the spatial distribution of wheat grain quality is provided in Figure 3, using the search terms “spatial pattern”, “underlying mechanism”, and “cultivation regulation”. Based on these matrices, protein, quality, starch, and nitrogen are the most researched topics. Although the spatial distribution of the chemical constituents in mature wheat grains has been reviewed thoroughly [14,15], questions remain regarding how these components are formed and how they react to cultivation practices. This article, therefore, first employs data from pearling and other approaches to summarize our recent knowledge of gradients in the mature wheat starchy endosperm and in developing caryopses, then discusses the underlying mechanism of spatial heterogeneous formation, and finally concludes with the regulatory effects of cultivation practices on quality spatial distribution.

## 2. Spatial Distribution of Grain Components

### 2.1. Protein

Kent analyzed the protein content of milled flour from the sub-aleurone layer and endosperm, respectively, and discovered that the protein content of the sub-aleurone layer was approximately two times higher than that of the endosperm [16]. With advances in milling technology, mature grains may now be ground layer by layer, yielding flour from bran, the aleurone layer, the sub-aleurone layer, and the outer, middle, and inner endosperm, respectively. This method may be used to investigate the spatial distribution of grain components, both systematically and comprehensively. Zhong et al. pearled grain into nine pearling fractions and assessed the protein content in each layer. The greatest amount was found in the outer endosperm layer and aleurone layer; the different wheat varieties showed a consistent trend [13]. The contents of two major storage protein components, gliadin and glutenin, display a unimodal pattern across the grain, reaching a peak at the outer layer of the grain, with the lowest level in the interior layer [13]. High molecular-weight glutenin subunit (HMW-GS) and glutenin macropolymer (GMP), which are positively correlated with bread strength, are also most abundant in the outer endosperm [13].

The distribution of protein in grains at the filling stage can be more intuitively observed by staining an ultrathin section, coupled with microscopy imaging. Xiong et al. observed the protein body (PB) distribution in 19-DAA caryopsis, revealing that the relative areas of PBs were higher in the ventral than in the dorsal region [17]. Based on ultrathin sections, George et al. divided the cross-section of the grain into 5 layers, from the outside layer to the inner, using ArcGIS software and quantified the PB area of each layer. Protein bodies with an area larger than 1 μm^2^ are more abundant in the outer layer of the endosperm, while those that are smaller than 1 μm^2^ are mostly concentrated in the inner endosperm [18]. Tosi et al. observed the spatial distribution of gluten protein caryopses using protein-specific antibody labeling combined with immunoelectron microscopy and discovered that the contents of γ-gliadin and HMW-GS are higher in the middle endosperm, while α-gliadin, ω-gliadin, and the low molecular-weight glutenin subunit (LMW-GS) are more abundant in the outer layer [19], which finding is consistent with the conclusion obtained by the combination of milling and biochemical analysis.

### 2.2. Starch

Starch consists of two types of glucose polymers: linear amylose, which is constituted of D-glucose linked by an α-(1, 4) glycosidic bond, and branching amylopectin with a high degree of polymerization, occupying 65–82% of the total starch [20]. Similar to the uneven distribution of protein in the grain, the amount and composition of starch varies across the grain. It was discovered that the quantities of total starch and amylose rose from the outer layer to the middle endosperm layer, after which they stayed essentially unaltered. The starch contents in the middle and inner endosperm were four times and three times higher than those in bran and aleurone, respectively. The amylopectin-to-amylose ratio declined dramatically from the wheat bran layer to the outer endosperm layer but did not change significantly from the outer endosperm layer inward [21].

The mature wheat grain contains primarily A-type (>10 μm) and B-type (<10 μm) starch granules [22]. The two types of starch granules were not uniformly distributed in the grains. Specifically, the number of B-type starch granules was greater in the sub-aleurone layer cells, whereas the number of A-type starch granules was greater in the middle endosperm and inner endosperm [23]. Zhou et al. discovered that the average particle size of starch granules in the different parts of the grain varied and that the average size of A-type starch granules increased from the outer layer to the inner layer [21]. Xiong et al. observed the section of caryopses 15 days after anthesis and found that the starch granules in the sub-aleurone layer were between 1.5 μm and 25.5 μm, with the peak values of B-type and A-type starch granules at 2.5 μm and 12.5 μm, respectively. Most of the starch granules in this area were compound, with an irregular shape. The size distribution of starch granules in the inner endosperm of the sample ranged from 1.5 μm to 39.5 μm, with peak values of 3.5 μm and 18.5 μm for B-type and A-type starch granules, respectively, and the predominant shape of the granules was observed to be regular and oval or spherical [17]. Tosi stained cross-sections of caryopses 20 days after anthesis and found that the number and particle size of the starch grains in the aleurone layer were significantly smaller than those in the endosperm layer, which finding was consistent with the spatial distribution of starch at the mature stage [24].

### 2.3. Other Components

Cell wall polysaccharides (CWPS) are the primary source of dietary fiber in cereal products. Wheat grains contain about 9.2% to 20.0% of total dietary fiber (TDF) [25]. The principal wheat dietary fiber fractions are non-starch polysaccharides, with mixed-linkage glucans and arabinoxylan (AX) being the predominant components in wheat grain [26]. β-glucans, which are mainly present in the inner aleurone cell walls and sub-aleurone endosperm cell walls, were found in a lower amount (0.4–0.8%), while the AX content was found to be relatively high in this cereal variety (0.5–8.8%) [27]. The content of cellulose in wheat ranges from 1.9% to 2.5% [28]. The grain bran includes nearly all the cellulose, AX, and β-glucan, but the white flour has less than 2% of the CWPS. The TDF content dropped from the outer layer to the inner layer, and the TDF, AX, and β-glucan levels in the inner endosperm were only 15.4%, 24.2%, and 31.5% of those in the bran [14]. Velickovic et al. employed MALDI mass spectrometry imaging (MSI) to map variations in the quantity and localization of polysaccharides in the endosperm [29,30]. It was found that AX and β-glucan were concentrated in the outer cells of the caryopses during grain filling and eventually moved inward as the grain matured. More recently, the same group observed the distribution of cell wall hemicelluloses in the wheat grain endosperm from a 3D perspective [31]. The result revealed that β-glucans were more prevalent in the central cells of the endosperm, while AX, and especially highly substituted AX, was more abundant in the cells surrounding the transfer cells. Fanuel et al. investigated the distribution of water and arabinoxylan structures in developing wheat grains, using magnetic resonance microimaging (MRI) in combination with MSI. They revealed for the first time that the distribution of water was spatially correlated with the acetylation and ARAF substitution of arabinoxylans [32]. A schematic model of the spatial gradients of protein, gluten, starch, and AX in wheat grain is shown in Figure 4.

Certain trace elements in wheat grains, such as mineral elements, vitamin B (mostly B9, B1, B2, B3, and B6), and phytin, impact both human health and the quality of flour processing [33]. Iron and zinc concentrations in wheat grain were found to decline steadily from the outer to the inner layer, whereas the ratios of soluble iron and zinc rose [34,35]. The Zn concentration in the aleurone layer can be up to 0.4%, while in the endosperm, it is typically lower than 0.01%, which is too low to meet dietary requirements of 0.03% [36,37]. More recently, Zhu et al. conducted a high-throughput metabolomic analysis and determined the contents and spatial distribution of metabolites in pearled fractions from the dried kernels of six representative bread-wheat varieties. The results showed that flavonoids varied the most among the cultivars and were found in higher concentrations in the outer layers of the grain, but were only at low concentrations in the starchy endosperm [38]. Due to the low amounts of other trace elements, it is only known that the level of vitamins or phytin is high in the bran and aleurone layer, but little is known about the spatial distribution of these compounds in the endosperm [5].

### 2.4. Processing and Baking Quality

The heterogeneous spatial distribution of chemical components such as starch and protein in grains inevitably results in spatial variances in the processing quality of flours from different layers. From the outer layer to the inner layer of the endosperm, the concentrations of total starch and amylose and the ratio of A-type starch granules rose significantly, resulting in an upward trend of flour gelatinization parameters including peak viscosity, trough viscosity, breakdown, final viscosity, setback value, and pasting time. In contrast, the gelatinization temperature decreased in the flour from the outer to the inner layers. The spatial variability of the storage protein and its components in different parts of the grain led to considerable changes in the farinograph parameters and gluten quality [39]. For instance, the alkaline water retention, sodium carbonate solvent retention, sucrose solvent retention, deionized water solvent retention, and lactic acid solvent retention of the flour initially increased and subsequently declined from the outer to the inner endosperm. High amounts of HMW-GS and GMP in the outer endosperm resulted in the highest levels of dry and wet gluten and the greatest dough strength [40,41]. In terms of the dough made from the outermost to the innermost flours, their developing time, stability time, extension area, and extensibility decreased, but the weakening degree and resistance to extension/extensibility increased [42].

The baking quality of flours from the different parts of the grain also showed significant spatial heterogeneity (Figure 5). Protein, dietary fiber, and mineral elements such as calcium, iron, and zinc were abundant in the bran and aleurone layer; however, the baked products of bran flour had a poor sensory evaluation because of its low starch content [13]. The volume, elasticity, and sensory score from bread baked using bran to using the inner endosperm rose at first but thereafter declined. The bread made using flour derived from the outer and middle endosperm was of the best quality, with relatively high cohesion and resilience and low hardness and chewiness [13]. The diameter of cookies made with successive layers of flour decreased at first and then increased. The thickness and hardness of the cookies decreased, while the sensory evaluation increased. The cookie created with inner endosperm flour has the greatest diameter and extension factor, the highest sensory quality score, the lowest thickness and hardness, and the finest baking quality among the cookies made with flour from different parts of the grain [13].

## 3. The Underlying Mechanisms of the Spatial Heterogeneity Formation of Protein and Starch in Wheat Grain

Wheat grain development can be divided into three stages: the cell differentiation and expansion stage (0–14 days after anthesis), when the structures of different parts of the grain are essentially formed and the synthesis of protein and starch begins; the grain-filling stage (14–28 days after anthesis), when the gluten and starch synthesize rapidly; and the dehydration stage (28 days after anthesis—maturity), when the grain filling rate slows and the water content decreases [43,44,45]. Generally, from 8 days after anthesis, starch biosynthesis is initiated in the amyloplast, with adenosine diphosphate glucose (ADPG) generated from the sucrose as a substrate, and by 10 days after anthesis, the gluten protein is synthesized on the rough endoplasmic reticulum [46,47,48]. Therefore, the formation of grain heterogeneity can be described using the following hypotheses: (1) protein and starch syntheses begin and end at different times, resulting in variable synthesis durations in different parts of the grain. For example, in maize, the programmed cell death (PCD) of the endosperm starts from the center of the endosperm and is carried out layer by layer in an orderly manner [49], so it takes longer for the outer endosperm to synthesize storage substances. However, Young et al. stained wheat grains with Evans blue and found that the PCD of wheat endosperm cells was disordered [50], indicating that the synthesis times of the same storage material in different parts of the grain were basically the same. More recently, it has been suggested that PCD in the starchy endosperm cells compromises storage product accumulation [51]. The PCD observed in the starchy endosperm occurs in a very specialized form in which only the cytoplasmic membrane loses integrity, while the membranes in most of the intracellular organelles, such as the nuclei, mitochondria, plastids, ER, and Golgi apparatus, remain functional and intact [52]. Taken together, the proposed hypothesis is not supported by the available evidence so far. (2) Spatial differences exist in the supply of sucrose or amino acid in different parts of the grain. (3) The differences of starch or protein synthesis capacity in different parts of grain lead to their spatial heterogeneity. This section is intended to provide an overview of hypotheses (2) and (3).

### 3.1. Substrate Supply and Spatial Heterogeneity of Protein and Starch in Wheat Grain

Ugalde and Jenner utilized microsections to investigate the composition of free amino acids and sugars in wheat grains at 20 days after anthesis. It was found that the gradients of amino acids and sucrose in caryopses did not correspond to the accumulation gradients of protein and starch, suggesting that the supply of heterogeneous substrates for grain protein or starch accumulation was not directly related [53,54]. Nevertheless, starch synthesis begins 5 days after anthesis, while storage protein synthesis initiates 10 days after anthesis [43], and the spatial heterogeneity of starch and protein begins to emerge 14 days after anthesis [19]. Therefore, it is necessary to analyze and clarify the changes in concentration and composition of free amino acids and sugars in different parts of the caryopses during the different grain-filling stages, particularly the dynamic spatial distribution at the early filling stage before protein and starch synthesis. However, due to the tiny size and high water content of caryopses at the early filling stage, it is difficult to gather samples from the different parts, and research in this area is proceeding slowly. Peukert et al. discovered that the contents of disaccharides, trisaccharides, and hexose in the inner endosperm, close to the cavity of the barley, were higher than those in the outer endosperm and aleurone layer, which was consistent with the spatial distribution of starch in mature grains [55]. Zhong et al. reported that the distribution pattern of free amino acids in the caryopses at the early filling stage (7–11 days after anthesis) was consistent with that of protein in the mature grain [56]. Therefore, the spatial heterogeneity of starch and protein distribution may be closely related to the substrate supply capacity in the early stage of grain filling. However, because of the rapid synthesis of starch and protein in the middle and late grain-filling stages, substrate consumption is extremely rapid. There is currently a paucity of direct data suggesting that the substrate supply capability over the whole filling phase causes the grain’s spatial heterogeneity of protein and starch. In addition, the relationship between the spatial distribution of lipids, dietary fiber, trace elements, and their synthetic substrates has not been reported.

The spatial difference in substrate supply capacity in grain is largely connected to its transport pathway. Wang et al. hypothesized that the substrate delivered by the spikelet vascular bundle is first emptied into the apoplast (endosperm cavity) before being transferred to the endosperm. Two transport pathways have been proposed: one is through the transfer cells surrounding the endosperm cavity, moving directly into the endosperm cells and being transported radially outward (transfer cell pathway); the other is transportation through the transfer cells into the aleurone layer, then transportation inwardly (aleurone pathway). Currently, the reports regarding these two transport routes are contentious [57]. Chen et al. observed the differences in protein body distribution in the grains using a resin-mounted semithin section via the Image-ProPlus software (Media Cybernetics, Rockville, MD, USA). The quantity and relative area of the protein bodies were determined to be as follows: abdominal outer layer > dorsal outer layer > abdominal inner layer, which may be related to the transport pathway of protein synthesis substrate amino acids. Since the distance between the endosperm cavity and the dorsal cells is not only longer than that between the endosperm cavity and abdominal cells but also requires more energy, it can be hypothesized that amino acids are delivered mostly via the aleurone pathway [58]. Similarly, Xiong et al. observed the accumulation dynamics of starch granules in caryopses and inferred that sucrose initially accumulated in the outer endosperm and was then transported to the inner endosperm, suggesting that starch synthesis substrates were also transported through the aleurone pathway [17]. However, Peukert et al. discovered that the expression and metabolic activity of genes involved in carbohydrate metabolism were greater in the cells surrounding the cavity in barley, and extrapolated that sucrose was transported predominantly via the transfer cell pathway [55]. As is consistent with their conclusion, Moore et al. injected 15N-labeled glutamine into wheat ears at the filling stage through a capillary and assessed the enrichment of 15N in the protein bodies using nano-secondary ion mass spectrometry (NanoSIMS). It was found that 15N-labeled glutamine was transported directly from the cavity to the endosperm via the transfer cells and, finally, to the sub-aleurone cells [59]. Taken together, there are still clear disputes in the findings of research on the substrate’s transport route, and prior studies could only deduce the substrate’s transport path by studying the spatial distribution of sugars or protein bodies. If the substrate movement throughout the caryopses can be visibly and noninvasively observed, it will be easier to determine whether the distribution of substrates limits the ability of protein starch synthesis in various areas.

### 3.2. Synthesis Capacity and Spatial Heterogeneity of Protein and Starch in Wheat Grains

Starch synthesis occurs primarily in the amyloplast, using sucrose as the starting point to catalyze the creation of the direct precursor of starch synthesis, ADPG, followed by the coordinated activity of a succession of starch synthases [60]. Gliadin and glutenin are storage proteins that undergo synthesis, folding, and deposition in the intimal system and accumulate within protein bodies [24]. The protein body generated in the early stage of grain development appeared in the form of a single ball with a diameter of approximately 2 mm. The single ball then expanded, gathered, and fused during the middle filling stage, then changed from spherical to irregular in shape. During the later filling stage, protein bodies were transformed into a protein matrix to fill the spaces between the starch granules [61]. Starch and protein synthesis is regulated by multiple genes. Previous researchers have hypothesized that the genes encoding storage proteins are mainly regulated at the transcriptional level [62], so there may be specific regulatory signals in the maternal tissues that regulate the transcription rate of genes encoding protein synthesis in different parts of the endosperm [19]. As a result of its proximity to the maternal tissue, the inner endosperm is more susceptible to disruption than the aleurone layer, and the expression level of genes encoding protein synthesis in the inner endosperm is relatively low. Ma et al. roughly divided grains into their inner endosperm part with a lower protein content and the remaining endosperm part with a higher protein content. The differentially expressed genes involved in carbon and nitrogen metabolism were determined to be mostly linked to amino acid conversion and transport. There was no difference in the expression of genes directly encoding protein synthesis between the two samples, indicating that the spatial distribution pattern of protein content may not be consistent with the ability to synthesize protein [63]. Moreover, Susan et al. discovered that the majority of genes involved in starch synthesis and degradation were up-regulated in the endosperm rather than in the aleurone layer, suggesting that the spatial distribution pattern of starch content may be connected to the expression level of the genes involved in starch synthesis [64]. However, the aforementioned study only split the grain into two portions, making it impossible to draw fair comparisons between the various parts of the grains.

The advancement of laser capture microdissection technology in recent years has enabled the precise collection of trace samples from the various regions of wheat grains. The pearling process described above may separate grains into distinct flour layers such as wheat bran, aleurone layer, outer endosperm, middle endosperm, and inner endosperm; however, it can only be used for mature grains and cannot be used to achieve pearled samples of fresh caryopses during grain filling. The laser capture microdissection technique may be used to monitor and find the target cell using a microscope, then mark the target region using a screen linked to the microscope, and then precisely cut the marked area with a laser to extract the target cell. The laser microdissection technique effectively overcomes the problem of cell heterogeneity by separating pure cell groups or even single cells from the fresh sample tissue [65]. In conjunction with a frozen section, it is possible to harvest high-quality RNA for transcriptome studies. Zhan et al. used this technique to isolate and collect five types of cells from the differentiated endosperm of maize for transcriptome analysis, identified the differentially expressed genes as maternal and offspring, and successfully established a model for the co-expression of different types of endosperm cell genes [66]. Zhong et al. effectively isolated the aleurone cells, outer endosperm, middle endosperm, inner endosperm, and transfer cells of caryopses at six filling stages and studied the expression levels of certain genes. It was revealed that the expression levels of the HMW-GS and gliadin coding genes were greater in the middle and inner endosperm, which was inconsistent with the protein spatial distribution pattern of the mature grain. Therefore, the spatial difference in protein distribution was not constrained by the ability of protein synthesis [56]. Due to the large number of genes involved in the process of protein and starch synthesis, it will be necessary in the future to obtain spatiotemporal genes from the different parts of grains at different filling stages in order to analyze, systematically and comprehensively, the relationship between protein and starch synthesis ability and grain quality spatial heterogeneity.

## 4. Spatial Heterogeneity of Grain Compositions as Affected by Cultivation Practices

### 4.1. Nitrogen Fertilizer

He et al. observed that the sulfur-poor gliadin concentration responded most substantially to nitrogen fertilizer increase and rose simultaneously in the whole grain. The raising effect of nitrogen fertilizer addition on the HMW-GS fraction of the outer endosperm layer was stronger than that on the inner endosperm layer, whereas the sulfur-rich gliadin showed a virtually inverse trend [11]. Similarly, Hermans et al. investigated the contribution of protein-rich sub-aleurone cells to protein content and gradient in the wheat endosperm of three cultivars under three levels of nitrogen fertilization, finding that N fertilization resulted in relatively higher increases in protein content in the outer compared to the inner endosperm [67]. The impact of nitrogen on starch quality was particularly visible in terms of variations in amylose content, the amylose to amylopectin ratio, starch granule distribution, and the pasting characteristics of flour from the middle and inner endosperm [39,68]. Zhong et al. recently reported that N application significantly increased the relative accumulation of beneficial bioactive substances (e.g., protein, amino acids, flavonoids, vitamins, and lipids) in the inner layer, but that excessive N application may inhibit this effect and lead to poor nutritional quality [15].

The timing of nitrogen topdressing also regulated the spatial distribution of the grain’s functional components. Delaying nitrogen topdressing timing from the jointing stage to the booting stage increased the gluten protein content in flour from all layers, but the rise was most pronounced in the aleurone and outer endosperm, and the regulation of the GMP and glutenin subunits was consistent with that of gluten protein. Shifting the timing of nitrogen topdressing from the jointing stage to the greening-up stage decreased the total protein and protein components, with the responses of the outer layer being more apparent [13]. Additionally, it had a direct impact on the spatial gradients of baking quality. Delaying nitrogen topdressing timing increased the volume and sensory assessment of bread cooked with flour generated from aleurone and the outer endosperm but had no influence on the baking quality of other layers. The effect of pre-dating nitrogen topdressing on bread baking quality (volume, appearance, and sensory evaluation) exhibited an opposing tendency.

### 4.2. Plant Density Combined with Nitrogen Availability

A cultivation practice of increasing density and decreasing nitrogen affected the spatial gradients of protein in soft wheat grain [69]. The spatial distribution of the soft wheat protein bodies was modulated by plant density combined with nitrogen availability, resulting in a change in the sedimentation value of the flour. The outer endosperm, followed by the middle and inner endosperms, responded to raising the density and lowering the nitrogen, in terms of grain protein quality. The simultaneous optimization of increasing density and nitrogen reduction allowed for the compensation of the yield loss brought on by nitrogen reduction [12], allowing the high yield and high quality of soft wheat to be synchronized. 

### 4.3. Sulfur Fertilizer and Amino Acids Spraying

The application of sulfur fertilizer increased the levels of total protein and glutenin in each grain layer, with the impact of the HMW subunit being stronger in the outer endosperm and middle endosperm, and gliadin concentration being higher in the outer endosperm than in the other layers [70]. Li et al. sprayed amino acid and urea onto two wheat varieties at anthesis and discovered that the number of A-type starch granules, average starch granule size, total protein, gliadin, glutenin and glutenin subunits, amino acid, gluten content, and sedimentation value all increased in all grain layers. Grain protein and its components, as well as glutenin subunits, first increased and then fell from the outside to the inner layer, with the GMP and amino acid levels rising higher in the outer than in the inner endosperm [71]. In contrast to the regulatory effects indicated above, Yang et al. discovered that spraying schemes of Met + Zn + Urea improved the protein quality of the inner endosperm rather than the aleurone layer more efficiently, resulting in the superior gluten quality and increased breadmaking quality of flour [72]. The regulatory effects of cultivation practices on the spatial gradients of protein in the wheat grain are summarized in Table 2.

### 4.4. The Spatial Heterogeneity in Grain Trace Elements as Affected by Cultivation Practices

In terms of trace elements, Shi et al. found that nitrogen application may enhance the contents of Fe and Zn in bran, but it was only the level of Zn that rose in terms of total flour [73]. According to Xue et al., the amount of Fe increased in wholemeal and wheat brans but dropped in refined flour [74], indicating that raising the concentration of Fe in flour via nitrogen application was difficult. In contrast, the addition of N fertilizer improved the concentrations and bioavailability of grain tissues, especially in the crease region and the endosperm [75]. Dong et al. observed that foliar spraying with zinc fertilizer at 5, 15, and 25 days after anthesis may successfully enhance the Zn content in different layers of grain, with the rise being larger in the outer layer of grain. Spraying zinc fertilizer, on the other hand, reduced the levels of Fe and Mn in wheat flour, with the mineral elements in the outer layer of grain being more impacted than those in the inner layer of grain [76]. Shao et al. also reported that Zn application significantly increased grain Zn content by 49.3%, with a maximum increase in the outer endosperm fraction [77].

## 5. Spatial Heterogeneity of Grain Compositions as Affected by Cultivation Practices

The spatial heterogeneity in grain composition affords the opportunity to manufacture specialty flour by optimizing the milling process. The flour from the middle and outer endosperm had a greater HMW-GS content and glutenin/gliadin ratio, as well as better dough strength, elasticity, and reduced viscosity, making it ideal for bread baking that demands a high gluten strength [78]. On the contrary, the inner endosperm layer has a greater starch content, a lower glutenin/gliadin ratio, and better dough extensibility, and is, hence, appropriate for baking cookies and related products. Shewry et al. pointed out that the flour from the outer endosperm has sufficient elasticity and strength to be used for making pasta; the difference in the ratio of amylose/amylopectin, combined with the difference in the contents of protein components, makes the flour from different parts of the grain also have potential applications in the production of quick-frozen food (such as steamed bread, bread, dumplings, etc.) [79]. Consequently, even when the protein or gluten level of the whole grain does not fit the traditional categories of “strong-gluten” or “weak-gluten” flours, the flour from selected layers can nonetheless meet the specific requirements of the final flour product.

Through the combining of multiple layers of flour based on their different qualitative characteristics, it is also feasible to make high-quality unique flour for specialized wheat products. Based on the flour derived from the middle endosperm and supplemented with flour from the aleurone layer, which is rich in protein and mineral elements, the combined flour exhibited decreased pasting properties and increased mineral element content. In addition, the baked bread became deeper in color, with increased volume, hardness, and chewiness. However, when the adding rate surpassed 20%, the bread’s quality and sensory ratings would fall dramatically. Adding bran layer flour that is based on endosperm flour lowered the combined flour’s pasting parameter, increased the mineral elements, and deepened the color of cooked cookies while decreasing the diameter, extension factor, and sensory score. The nutritional quality and taste of the cookies improved when the second layer of wheat grain flour was blended with the fifth layer of flour at a 2:8 ratio, or when the first layer was blended with the ninth layer at a 1:9 ratio [80].

## 6. Perspectives

### 6.1. Technological Advancements Facilitate the Mechanism Investigation of Spatial Heterogeneity Formation

To summarize, the spatial heterogeneity of key components in wheat grain has generally been obvious; however, research on the mechanism of this spatial heterogeneity is hampered owing to sampling difficulties, tiny caryopses size, the huge wheat genome, and other considerations. Recently, the development of nondestructive imaging tools that observe substrate transport and allocation in grains on the appropriate spatial and temporal scales facilitate transport pathway investigations. For instance, positron emission tomography-computed tomography (PET-CT) has been developed to offer a practical imaging technique that is used to assess the entire plant transport and nutrient allocation by quantifying the distribution of positron-emitting radioisotopes in a plant non-invasively and over time [81,82,83]. Currently, PET-CT has been applied to numerous plant studies, focusing on the transport and allocation of photoassimilates in sorghum [84], giant reed [85], and tobacco [86]. 

### 6.2. Directional Regulation of Key Gene Expression Modifies the Spatial Distribution Pattern

Furthermore, the advancement of laser microdissection technology and multi-omics analyses will aid in the investigation of the mechanisms behind spatial heterogeneity formation. In the future, the aforementioned technology can be used to obtain cells from any part of the fresh caryopses in conjunction with multi-omics methods to clarify the key regulatory processes, key regulatory genes, or gene networks responsible for the formation of spatial heterogeneity in the context of grain quality. Tauris et al. employed laser microdissection technology to extract the barley grain aleurone layer, nucellus process, embryo, and endosperm cells, which they then paired with microarray technology to determine the zinc transport pathway in the grain to reveal a key zinc transporter (HvMTP1) regulating zinc distribution in barley grains [87]. After that, the researchers specifically overexpressed *HvMTP1* in endosperm cells, which successfully enhanced the endosperm zinc concentration and improved zinc transport and transfer from the aleurone layer to the endosperm [88]. This establishes a precedent for the regulation of the spatial heterogeneity of wheat grain components, showing that the spatial heterogeneity may be modified by directionally changing the expression of important genes. 

### 6.3. Differential Spatial Responses to Cultivation Practices Have the Potential for Specialized Wheat Production

Grain protein accumulation responds to nitrogen application differentially, as do the spatial gradients of the functional components in wheat grain [11,13,15]. Certain genotypes are more susceptible to nitrogen in their outer layers, while others are more sensitive to it in their interior layers. This implies that the response of composition gradients to nitrogen fertilizer application differs according to genotype. Therefore, it will be feasible to produce specialized wheat that is both high-yielding and of high quality if the essential genes governing spatial gradient formation in the grains and reactions to nitrogen fertilization can be accurately regulated.

## Figures and Tables

**Figure 1 plants-12-02192-f001:**
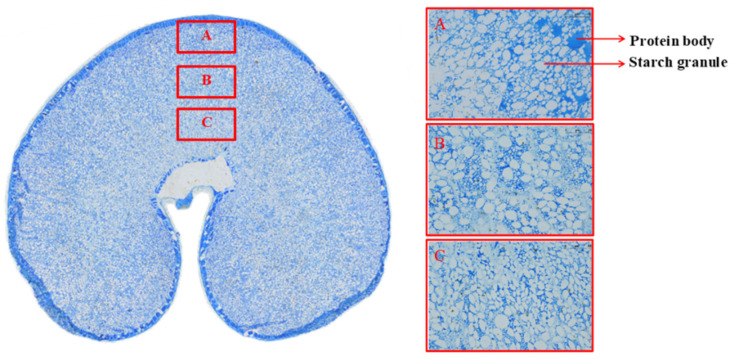
Cross-section of a developing grain stained with Coomassie bright blue G250. Note: A mature grain of Ningmai 13, sliced at a thickness of 2 μm. A, B, C represents the areas of outer endosperm, middle endosperm and inner endosperm, respectively.

**Figure 2 plants-12-02192-f002:**
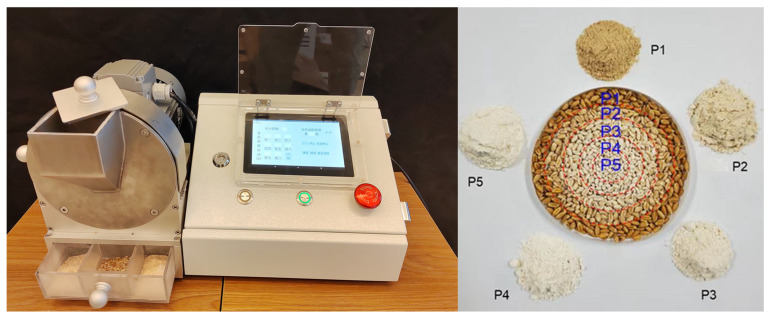
Flour from different pearling fractions, milled using a grain pearling machine (Huitong, China). Note: P1 to P5 indicates a range of pearling from fraction 1 to fraction 5, milled from the outer to the inner grain parts, respectively.

**Figure 3 plants-12-02192-f003:**
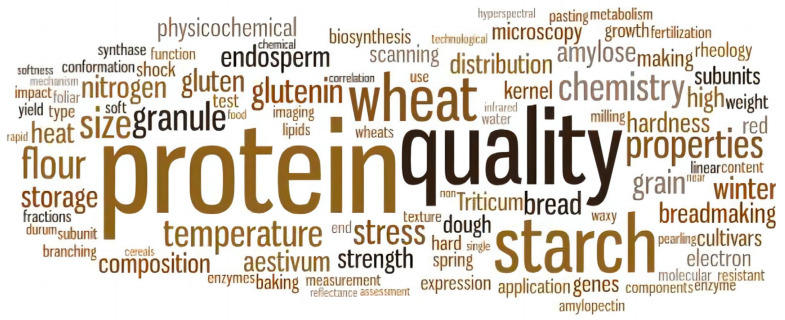
Word cloud showing the frequency of terms in scientific papers on the spatial gradients of wheat grain. Source: Web of Science; search term: “(spatial pattern AND (underlying mechanism OR cultivation regulation) AND wheat”. Search conducted: 16 May 2023. The abstracts from 63 artificially screened hits were used to generate the word cloud using wordle.net (the number of times a word appears determines its relative size).

**Figure 4 plants-12-02192-f004:**
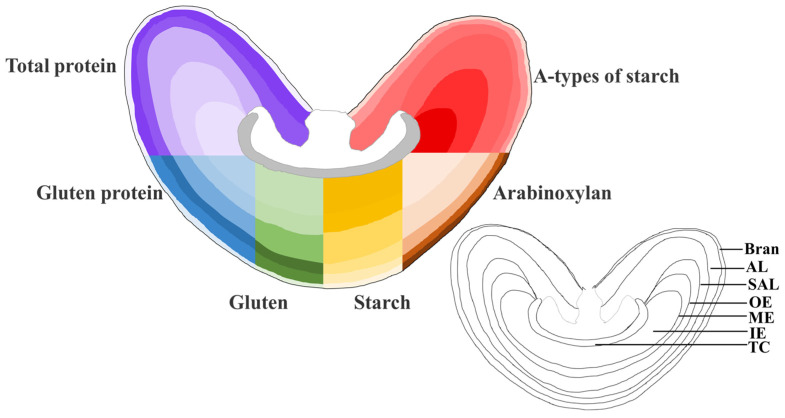
Schematic model of the spatial gradients of protein, gluten, starch, and arabinoxylan in wheat grains [13,14,21]. Note: AL, aleurone layer. SAL, subaleurone layer. OE, outer endosperm. ME, middle endosperm. IE, inner endosperm. TC, transfer cells. The content of each component is positively correlated with the corresponding color depth.

**Figure 5 plants-12-02192-f005:**
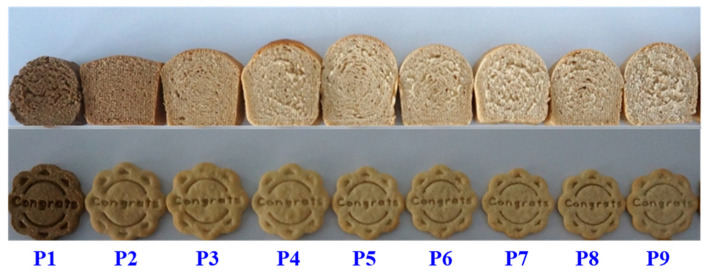
Appearance and texture of bread and cookies made of different pearling fractions of wheat grain [21]. Note: P1 to P9 indicates pearling fraction 1 to fraction 9, from the outer to the inner grain parts, respectively.

**Table 1 plants-12-02192-t001:** Quality-grading criteria for wheat in Australia, China, the United States, and Russia.

Country	Australia											China				
Classification of Wheat	Prime Hard Wheat	Hard Wheat	Premium White Wheat		Standard White Wheat		Standard Noodle Wheat		Premium White Wheat—Noodle Wheat	Soft Wheat	Durum Wheat	Strong Gluten Wheat	Medium–Strong Gluten Wheat	Medium Gluten Wheat		Weak Gluten Wheat
Protein content (%)	≥13.0	≥11.5	≥10.0		-		9.5–11.5		10.0–11.5	<9.5	>13.0	≥14	13.0–14.0	12.5–13.0		<12.5
Bulk density(g/L)	≥760	≥760	≥760		≥760		≥720		≥760	-	-	≥770	≥770	≥770		≥750
Drop value(s)	≥350	≥300	≥300		≥300		≥300		≥300	-	-	No data source				
Gluten content (%)	No data source											≥30	28–30	25–28		<25
Data sources	https://wheatquality.com.au/ (accessed on 22 February 2023) https://www.aegic.org.au/australian-grains/wheat/ (accessed on 22 February 2023) https://www.graintrade.org.au/ (accessed on 22 February 2023)											Quality classification of wheat varieties GB/T17320-2013				
**Country**	**United States**						**Russia**									
**Classification of Wheat**	**Hard Red Winter**	**Hard Red Spring**	**Hard White**	**Durum**	**Soft White**	**Soft Red Winter**	**Durum Wheat**					**Soft Wheat**				
							**Grade I**	**Grade II**	**Grade III**	**Grade IV**	**Grade V**	**Grade I**	**Grade II**	**Grade III**	**Grade IV**	**Grade V**
Protein content (%)	10.0–13.0	12.0–15.0	10.5–14.0	12.0–15.0	8.5–10.5	8.5–10.5	≥13.5	≥12.5	≥11.5	≥10.0	-	≥14.5	≥13.5	≥12.0	≥10.0	-
Bulk density(g/L)	No data source						≥770	≥770	≥745	≥710	-	≥750	≥750	≥730	≥710	-
Drop value(s)	No data source						≥200	≥200	≥150	≥80	-	≥200	≥200	≥150	≥80	-
Gluten content (%)	No data source						≥28	≥25	≥22	≥18	-	≥32	≥28	≥23	≥18	-
Data sources	Overview of U.S. WHEAT INSPECTION (2021). https://www.uswheat.org/wp-content/uploads/Overview-of-U.S.-Wheat-Inspection.pdf (accessed on 22 February 2023)						Russian national standard ΓOCT9353-2016									

**Table 2 plants-12-02192-t002:** The regulatory effects of cultivation practices on protein gradients in wheat grain.

Cultivation Practice	Regulatory Effect			
	Aleurone	Outer Endosperm	Middle Endosperm	Inner Endosperm
Increase N level	+++	++	++	+
Delay N topdressing timing	+++	+++	++	+
Increase planting density and decrease N level	+++	+++	++	+
Apply S fertilizer	++	+++	+++	+
Foliar spray of Zn fertilizer	++	+	+	+++
Foliar spray of amino acid	++	++	+++	+

Note: the number of ‘+’ symbols represents the significance of regulatory effects on the protein contents of different positions in wheat grain under the same cultivation practice.

## Data Availability

All data cited in the study are publicly available.

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
