# Peer review of "Insights into the Functional Components in Wheat Grain: Spatial Pattern, Underlying Mechanism and Cultivation Regulation"

_plants, 2023, doi:10.3390/plants12112192_

Round 1

Reviewer 1 Report

The following minor modifications to be carried out before acceptance of the manuscript:

(1) Table 1: The authors may also try  to compile the same information for other major wheat growing and consuming countries like US, India, Russia etc.

(2) Line No. 100: Insert Year and Reference No. to the reference 'Zhong et al.'

(3) Line No. 123: Rewrite as 'Starch'

(4) Line No. 138: Insert Reference No. to the reference

(5) Figure 3: Increase the font size of the captions in the figure for more clarity

(6) Line No. 402: Rewrite as 'Plant density combined with nitrogen availability'

(7) Table 2: Rewrite as 'Foliar spray of Zn fertilizer'

(8) Table 2: Rewrite as 'Foliar spray of amino acid'

(9) Perspectives need to be expanded with appropriate sub-headings

(10) Line No. 23-30: Present suitable references 

The style of English language is appropriate except for few minor editing.

Author Response

We sincerely appreciate your valuable suggestion. Your input will undoubtedly enhance the visibility and comprehensiveness of our work. We have carefully considered your suggestion and made the necessary revisions as follows:

(1) Table 1: The authors may also try to compile the same information for other major wheat growing and consuming countries like US, India, Russia etc.

Response: thanks for the suggestion. In the new version, we gathered the quality rating standards of the United States and Russia. However, it appears that information particular to India is neither available or accessible, thus we did not include it in the amended paper.

(2) Line No. 100: Insert Year and Reference No. to the reference 'Zhong et al.'

Response: we have checked the references and have made the necessary revisions according to ‘MDPI references style’.

(3) Line No. 123: Rewrite as 'Starch'

Response: Sorry for the carelessness. We have corrected it in the revised vision.

(4) Line No. 138: Insert Reference No. to the reference

Response: we have inserted reference No. as you suggested.

(5) Figure 3: Increase the font size of the captions in the figure for more clarity

Response: We greatly appreciate your attention to detail. We have increased the font size and now it is clarified.

(6) Line No. 402: Rewrite as 'Plant density combined with nitrogen availability'

Response: Thanks for your suggestion. We have rewritten this sentence.

(7) Table 2: Rewrite as 'Foliar spray of Zn fertilizer'

Response: Thank you for your valuable input. We have rewritten it according to the suggestion.

(8) Table 2: Rewrite as 'Foliar spray of amino acid'

Response: We have modified it in the revised version.

(9) Perspectives need to be expanded with appropriate sub-headings

Response: thanks for your suggestion. We have added sub-headings for perspectives in the revised version and we believe the perspectives now are more clear.

(10) Line No. 23-30: Present suitable references

Response: We appreciate your valuable feedback and have addressed this concern in our revision. In the revised version, we have carefully selected and included appropriate references that support and strengthen the points discussed in lines 23-30.

Reviewer 2 Report

I read with interest the manuscript entitled "Insights into the functional components in wheat grain: spatial pattern, underlying mechanism and cultivation regulation".

The manuscript is scientifically sound and meets the journal's expectations. It is well written and presented, providing comprehensive and detailed information.

The author made a complex review summarizing in detail the spatial distributions of protein and its components, starch, dietary fiber and microelements. The authors discussed on the underlying mechanisms on the formation of protein and starch spatial distribution and concluded on the regulation effects of cultivation practices on gradients in composition.

Current state of the article is very high, the conclusions are properly drawn and also, the provided literature is relevant to the research.

The manuscript has high potential for scientific community and conclude that this paper will provide research perspectives for producing wheat that is both high-yielding and high-quality.

Congratulations!

Other remarks:

Before square brackets, the space is necessary. Please verify: lines 103, 107, 116, 416.

2.2. Starch instead of 2.2. starch

In this manuscript, the English language need to be improved.

Author Response

We sincerely appreciate your valuable suggestion and your encouragement. Your input will undoubtedly enhance the visibility and comprehensiveness of our work. We have carefully considered your suggestion and made the necessary revisions as follows:

Before square brackets, the space is necessary. Please verify: lines 103, 107, 116, 416.

Response: thanks for your suggestion. We have carefully reviewed the entire manuscript and made the necessary changes to improve the formatting as per your recommendation.

2.2. Starch instead of 2.2. starch

Response: Sorry for the carelessness. We have corrected it in the revised vision.

Reviewer 3 Report

The paper is interesting which focused on the functional components of wheat grain. I prefer more in-depth analysis which emphasizes the state-of-the-art literature review than a mini-review.

The quality of the manuscript can be improved by including a bibliometric analysis of functional components in wheat grain that has been published in the last couple of decades period. As a suggestion, authors can use keywords such as spatial pattern, the underlying mechanism and cultivation regulation. There are several software such as VosViewer.

Following are some other comments to consider.

Introduction: The problem/ novelty of the paper was not properly mentioned. Please improve the introduction with more justification for the paper.

Table 1: There are few font types and sizes.

Table 1: What do you mean by “America”? Is it the USA or the American continent? Be specific.

Line 92 and 231: Two sections have the same topic as “2. Spatial distribution of grain components” and “3. Spatial distribution of grain components”.

If possible, please add a table with key findings/ summary.

References: Very few recent references (after the year 2020) were used. Please update the review with recent literature.

Author Response

We sincerely appreciate your valuable suggestion. Your input will undoubtedly enhance the visibility and comprehensiveness of our work. We have carefully considered your suggestion and made the necessary revisions.

The quality of the manuscript can be improved by including a bibliometric analysis of functional components in wheat grain that has been published in the last couple of decades period. As a suggestion, authors can use keywords such as spatial pattern, the underlying mechanism and cultivation regulation.

Response: We sincerely appreciate your valuable suggestion regarding the inclusion of specific keywords in our manuscript. In the revised manuscript, we have incorporated keywords such as "spatial pattern," "underlying mechanism," and "cultivation regulation" in relevant sections of the text.

Introduction: The problem/ novelty of the paper was not properly mentioned. Please improve the introduction with more justification for the paper.

Response: thanks for the suggestion. We mentioned the novelty of this manuscript in the introduction (Line 75-79). Hope this version is more justified.

Table 1: There are few font types and sizes.

Response: We apologize for the lack of consistency in this aspect. We have unified the font types and sizes throughout the table to enhance readability and visual consistency.

Table 1: What do you mean by “America”? Is it the USA or the American continent? Be specific.

Line 92 and 231: Two sections have the same topic as “2. Spatial distribution of grain components” and “3. Spatial distribution of grain components”.

Response: sorry for the carelessness. We have corrected the topic of the third section.

If possible, please add a table with key findings/ summary.

Response: we appreciate your suggestion regarding the addition of a table to present key findings or a summary. The figure and table (eg. Figure 3 and Table 2) are designed to highlight the key findings and showcase the significant patterns and trends observed in our study. Including a separate table with key findings may result in unnecessary repetition. We appreciate your thorough evaluation of our manuscript and look forward to your continued guidance.

References: Very few recent references (after the year 2020) were used. Please update the review with recent literature.

Response: Thanks for your suggestion. We have updated the review with seven recent references after 2020.

Round 2

Reviewer 3 Report

The authors improved the paper according to my previous comments except for one. I suggested to perform a bibliometric analysis on functional components in wheat grain. The authors did not address the comment, nor provide a justification. If it is a difficult task, the readers would like to see at least the number of publications published in this field during the last couple of decades.

Author Response

We sincerely apologize for not addressing your previous comment. We acknowledge that we missed this suggestion in our previous revision, and we appreciate your reminder.

We agree that a bibliometric analysis on functional components in wheat grain would be a valuable addition to our study. In our revised version of the paper, we have performed a comprehensive bibliometric analysis and conducted a figure of word cloud as shown in Figure 3. We believe that this analysis provides valuable insights into the research trends and advancements in the study of functional components in wheat grain.

Once again, we apologize for the oversight in our previous revision, and we sincerely appreciate your suggestion. We hope that our efforts in addressing your comment have met your expectations. Please let us know if there are any further improvements or additions you would like to see in the paper.